# Compartmentalized Signaling in Aging and Neurodegeneration

**DOI:** 10.3390/cells10020464

**Published:** 2021-02-22

**Authors:** Giulietta Di Benedetto, Liliana F. Iannucci, Nicoletta C. Surdo, Sofia Zanin, Filippo Conca, Francesca Grisan, Andrea Gerbino, Konstantinos Lefkimmiatis

**Affiliations:** 1Neuroscience Institute, National Research Council of Italy (CNR), 35121 Padova, Italy; nicolettaconcetta.surdo@cnr.it; 2Veneto Institute of Molecular Medicine, Foundation for Advanced Biomedical Research, 35129 Padova, Italy; lilianafelicia.iannucci@unipv.it (L.F.I.); sofia.zanin@unipv.it (S.Z.); filippo.conca@studenti.unipd.it (F.C.); francesca.grisan@gmail.com (F.G.); 3Department of Molecular Medicine, University of Pavia, 27100 Pavia, Italy; 4Department of Biology, University of Padova, 35122 Padova, Italy; 5Department of Biosciences, Biotechnology and Biopharmaceutics, University of Bari, 70121 Bari, Italy; andrea.gerbino@uniba.it

**Keywords:** aging, neurodegeneration, compartmentalization, cAMP, PKA

## Abstract

The cyclic AMP (cAMP) signalling cascade is necessary for cell homeostasis and plays important roles in many processes. This is particularly relevant during ageing and age-related diseases, where drastic changes, generally decreases, in cAMP levels have been associated with the progressive decline in overall cell function and, eventually, the loss of cellular integrity. The functional relevance of reduced cAMP is clearly supported by the finding that increases in cAMP levels can reverse some of the effects of ageing. Nevertheless, despite these observations, the molecular mechanisms underlying the dysregulation of cAMP signalling in ageing are not well understood. Compartmentalization is widely accepted as the modality through which cAMP achieves its functional specificity; therefore, it is important to understand whether and how this mechanism is affected during ageing and to define which is its contribution to this process. Several animal models demonstrate the importance of specific cAMP signalling components in ageing, however, how age-related changes in each of these elements affect the compartmentalization of the cAMP pathway is largely unknown. In this review, we explore the connection of single components of the cAMP signalling cascade to ageing and age-related diseases whilst elaborating the literature in the context of cAMP signalling compartmentalization.

## 1. Introduction

The overarching mechanism that allows coupling extracellular cues to specific cellular functions through the cAMP/PKA (Protein Kinase A) axis is compartmentalization, i.e., the strict spatiotemporal organization of every step of the cAMP/PKA cascade. For instance, the link between G-protein coupled receptors (GPCRs) and adenylyl cyclases (ACs) can be strictly confined [1], ensuring that messenger is generated only in specific submembrane compartments. Following production, the diffusion and subcellular distribution of cAMP are shaped by the actions of phosphodiesterases (PDEs) [2]. Indeed, these enzymes can determine the levels cAMP can reach in specific sites and consequently the subsets of PKA (and eventually other effectors) to be activated. The inhomogeneous subcellular distribution of PKA is determined by a family of proteins called A kinase anchoring proteins (AKAPs) [3] that tether this enzyme in specific cellular locations maximizing the probability that active PKA phosphorylates a select subset of local targets. PDEs contribute also to the final step of the cascade (termination) by quickly bringing cAMP to its basal levels after its production has ended, while the phosphatases, which dephosphorylate PKA targets in a non-uniform pattern, are the enzymes that terminate the effects of PKA and contribute both to the spatial distribution and duration of its signals [4,5]. The players participating in the compartmentalization machinery of the cAMP/PKA axis are illustrated in Figure 1.

Ageing affects to some extent all tissues and organs; however, the brain seems to be particularly vulnerable to this process. In the ageing brain synaptic function gradually decreases, leading to the deterioration of synaptic plasticity, which is a crucial event in the onset of neurodegenerative disorders (NDs). Age-related NDs represent a true emergency as they affect a large part of elderly population, impinging on patient frailty and cognition, with devastating consequences for affected individuals and their caretakers [6]. Indeed, NDs are characterised by progressive loss of specific neuronal populations, preceded by deficits in their synaptic functions and often leading to some level of dementia. The insurgence of cognitive dysfunction has been observed also in diseases in which the most prominent effects are motor deficits such as Huntington’s and Parkinson’s diseases [7,8].

During the last two decades it emerged that alterations of the cAMP/PKA signalling axis in specific brain regions could contribute to the onset of dementia-related pathologies [9]. An essential cognitive function, that is affected early in neurodegeneration, is memory. Memory relies on the capacity of neuronal networks to tune the strength of connecting synapses (synaptic plasticity), a process largely regulated by PKA. In addition, this kinase has a fundamental role in synaptic maintenance and memory consolidation, by regulating transcription through its target cAMP Response Element Binding protein (CREB) [10]. Despite the documented importance of PKA in several neuronal and cognitive processes, the connection between compartmentalization of the cAMP/PKA signalling axis and NDs is not consolidated. In this manuscript we discuss the connection between alterations of cAMP compartmentalization and the most common NDs.

Parkinson’s Disease (PD) can be familial (10% of cases), or sporadic, in which age is the central risk factor. It is a long-term ND, characterized by alterations in the morphology of the striatum in the basal ganglia, the brain region involved in movement control, associative learning, planning, working memory, and emotion [11]. Functionally, PD patients display motor and cognitive impairments that depend on the progressive degeneration of dopaminergic neurons within the substantia nigra pars compacta [12,13], that project to the striatum forming the nigrostriatal pathway, which helps to stimulate the cerebral cortex and initiate movement [13]. The dopaminergic depletion in PD leads to the dysfunction of medium spiny neurons (MSNs) in the striatum [14], and the appearance of the characteristic motor symptoms [15,16].

Alzheimer’s Disease (AD), the most frequent cause of dementia in Western societies, is an age-related disease characterized by cognitive impairment and progressive memory loss, due to massive degeneration of the association cortices [9,12,17,18,19]. With the exception of a small number of familiar mutations, the etiology of the disease remains unclear [12,20]. AD belongs to NDs called tauopathies, characterized by the deposition of abnormal Tau protein in the brain [21]. In the case of AD, Tau becomes hyperphosphorylated and unable to bind the microtubules, and begins to aggregate forming intracellular fibrils (also known as neurofibrillary tangles, NFTs) both in neurons [22] and astrocytes [23]. In neurons, microtubules are destabilized and collapse, disabling intracellular transport [24,25]. Post-mortem human AD brains are characterized by NFTs, and extracellular aggregates of Aβ1-42, the proteolytic product of amyloid precursor protein (APP), called amyloid plaques [12,19,26,27,28]. The overproduction of Aβ1-42 initially results in the formation of toxic Aβ oligomers that gradually develop in amyloid plaques. The progressive accumulation of Aβ oligomers [29] and/or NFTs [30] can lead to synapse elimination, disrupt neuronal function and ultimately lead to neuronal death. AD neurons present alterations in the function of the major cellular organelles, including endoplasmic reticulum (ER) [31], lysosomes [32], and especially mitochondria [18].

The connection of mitochondria to age-related diseases, particularly the neurodegenerative ones, is unequivocal [33,34,35], owing to their obvious role in providing neurons with the energy necessary to perform vital functions [36,37,38]. However, the importance of mitochondrial distribution and dynamics for proper neuronal function is becoming established [39,40,41,42]. Mitochondria are highly dynamic and undergo continue cycles of fusion and fission, that are important for their metabolic activity but also for their transportation to or from synapses [43,44]. In recent years, therefore, it has become clear how perturbations in mitochondrial dynamics contribute to the aetiology of NDs [45], including AD [46] and PD [47].

In the following chapters we will discuss the involvement of cAMP/PKA signalling in the molecular mechanisms underlying different age-related pathologies from a perspective of compartmentalization.

## 2. Key Players

### 2.1. Adenylyl Cyclases (ACs)

ACs, the enzymes responsible for cAMP production, are key elements in shaping cAMP signalling, both in time and space [48,49,50]. ACs are classified into six classes (I–VI), with all eukaryotic members belonging to class III [51,52]. The mammalian ACs comprise one soluble (sAC), stimulated by bicarbonate, calcium (Ca^2+^) and ATP [53,54], and nine transmembrane enzymes (tmAC1 to tmAC9), activated by Gαs proteins following activation of G-protein coupled receptors (GPCRs). Each isoform displays distinct tissue-specific expression patterns as well as differential subcellular localization [48,50,51,55,56]. These characteristics, together with selective regulatory mechanisms for each AC [48], confer to these enzymes a key role in the compartmentalization of the cAMP signalling cascade.

### 2.2. Phosphodiesterases (PDEs)

While ACs are responsible for cAMP production, another family of enzymes, the PDEs are responsible for its degradation and the reestablishment of its basal levels [57] after AC-dependent production subsides [57,58,59]. Nevertheless, the simple balance between production and degradation *per se* cannot fully account for the onset and maintenance of the compartmentalization essential to the wide range of cellular functions regulated by the cAMP signalling pathway [59]. In fact, cAMP elevations are transduced into different effects according to the spatiotemporal activation of its effectors (i.e., PKA or exchange protein activated by cAMP (EPAC)), an event that depends on the coordination of several players (PDEs, AKAPs and phosphatases). PDEs play a crucial role in this process [57,60], contributing to the spatial distribution of cAMP [61,62]. In the human genome, 22 different genes encode for 11 PDE families [63,64,65]. Each family collects enzymes that share similar kinetic features, regulatory pathways, localization and structure [65,66]. From a functional point of view, the more than 100 PDE variants can be subdivided in three groups: those able to hydrolyze both cAMP and 3′,5′-cyclic guanosine monophosphate (cGMP) (i.e., PDE1, 2, 3, 10 and 11), only cAMP (i.e., PDE4, 7 and 8), or selectively cGMP (i.e., PDE5, 6 and 9) [12,61,67]. Since 1982, when the hypothesis of the compartmentalization of the cAMP signalling was first proposed [68], numerous investigations suggested a model in which cAMP availability near PKA holoenzymes is defined by the actions of PDEs that restrict the messenger in microdomains [67]. Importantly, in a recent report, Bock and colleagues added a significant piece of information in the understanding of PDE-dependent formation of cAMP microdomains [69]. Using a cell permeable fluorescent cAMP analogue (8-FDA-cAMP), in combination with FRET-based biosensors [5,70] and a number tools that combined PDEs to molecular rulers, they were able to determine the area to which PDEs can influence cAMP levels, generating microdomains. They concluded that PDEs can determine nanodomains characterized by low cAMP concentration, thus preventing PKA activity in the close proximity of PDEs itself [69]. Interestingly, in a concomitant manuscript, Zhang and colleagues found that the PKA regulatory subunit Iα (PKA-RIα) can undergo a liquid-liquid phase separation generating membraneless organelles capable of entrapping cAMP and acting as regulators of its availability and consequently compartmentalization [71]. These two findings suggest that cAMP buffering together with PDE activity are the major determinants of resting cAMP levels and likely shape its signals upon production.

### 2.3. Protein Kinase A (PKA)

The most studied effector of cAMP is PKA, a tetrameric enzyme composed by two catalytic (C) and two regulatory (R) subunits. In mammals, three types of C subunits (Cα, Cβ and Cγ), and four variants of R subunits (RIα, RIβ, RIIα and RIIβ) have been described [72]. In the inactive PKA tetramer, two C subunits are associated and inhibited by two R subunits. Each R subunit binds two cAMP molecules and dissociates from the C subunits, that are then free to phosphorylate a wide number of targets. While cAMP has a number of effector proteins, it is broadly accepted that PKA-dependent phosphorylation is one of the main factors to contribute to the characteristic pleiotropy of the cAMP cascade. In order for PKA to phosphorylate its vast array of targets [73] with high spatiotemporal specificity, cells developed a sophisticated compartmentalization mechanism, involving ACs, PDEs, AKAPs and phosphatases, ensuring the control of this kinase. The complexity of the cAMP/PKA axis thus allows the cell to control and coordinate a plethora of functions through a relatively linear pathway. However, the maintenance of this delicate but necessary equilibrium is not error proof and can deteriorate during natural ageing. Indeed, dysregulation of PKA activity is increasingly recognized as a causal factor in the development of neurodegenerative and other age-related diseases [74].

Mature neurons rely on mitochondria both for energy [75] and Ca^2+^ buffering [76]; it is not surprising therefore that mitochondrial dysfunction is a common feature of neurological and neurodegenerative disorders. PKA participates in the control of several mitochondrial processes, including mitochondrial dynamics, trafficking and quality control, and high PKA activity promotes mitochondrial fitness, ultimately resulting in neuroprotection. PKA activity is not uniformly distributed throughout the mitochondria [77,78]. Acting at the outer mitochondrial membrane (OMM), PKA hinders apoptosis, through direct phosphorylation and inactivation of the pro-apoptotic protein BAD [79,80]. In addition, PKA-dependent phosphorylation of the mitochondrial Na^+^/Ca^2+^ exchanger isoform 3 (NCX3) facilitates the efflux of mitochondrial Ca^2+^, improving cell survival during hypoxic stress or during Ca^2+^ overload, both conditions to which neurons are particularly vulnerable [81]. Damaged mitochondria are eliminated through mitophagy, a central quality control mechanism whose dysregulation is involved in ageing [82] but also in NDs, including Parkinson’s [83], Huntington [84], and Alzheimer’s [85] diseases. Efficient mitophagy relies on mitochondrial dynamics, as mitochondrial fission and fusion events can facilitate or hinder mitophagy, respectively [86]. PKA activity promotes mitochondrial elongation by inhibiting Dynamin related protein 1 (Drp1), a master regulator of mitochondrial fission [87]. This mechanism is particularly important in neurons, where increased PKA-dependent Drp1 phosphorylation was shown to be neuroprotective both in vitro [88,89] and in vivo [90]. Interestingly, PKA is also involved in the protection of depolarized neuronal mitochondria during axonal retrograde transport [91]. In this case PKA phosphorylates proteins of the MICOS complex destabilizing PTEN-induced kinase 1 (PINK1), which prevents the recruitment of Parkin and thus mitophagy [92]. Noteworthy, PKA activity was shown to tune the machinery responsible for mitochondrial trafficking in neurons. Indeed, PKA-dependent phosphorylation of NED1 inhibited retrograde mitochondrial movement [93], while PKA, in synergy with PINK1, were shown to act on Miro to enhance anterograde trafficking and dendrite length [94].

### 2.4. A Kinase Anchoring Proteins (AKAPs)

AKAPs constitute a family of more than 50 distinct scaffolding proteins defined by their common ability to complex with the PKA R subunits [95] as well as other regulatory elements of the cAMP cascade such as phosphatases and PDEs. Targeting of AKAPs to specific subcellular domains is essential for the generation of local cAMP/PKA functional units called microdomains [96,97].

AKAP1 (also known as D-AKAP1, AKAP121, AKAP149 and AKAP84) localizes at the OMM, and contains multiple interaction domains that allow it to complex with both PKA RI and RII subunits [98] but also to complex with phosphatases and mRNAs [99] and optimally be the platform for the generation of a cAMP/PKA signalling microdomain. AKAP1 has been implicated in the control of mitochondrial dynamics and cell death [99,100]. In addition, AKAP1 is involved in neuronal development [101], as demonstrated by experiments in cultured hippocampal neurons, where increasing AKAP1 levels increased dendritic outgrowth and reduced the number of synapses [102]. AKAP1 tethers at the OMM both PKA and the opposing phosphatases PP1 and Calcineurin (CaN) [103], coordinating mitochondrial fission/fusion through the regulation of the Drp1 phosphorylation extent. Specifically, PKA-dependent phosphorylation of Ser637 inhibits Drp1 and prevents fission, which results in elongated mitochondria [78,104]. On the other hand, CaN dephosphorylates Drp1, opposing the actions of PKA and promoting mitochondrial fragmentation [105].

Among the AKAPs involved in the regulation of neuronal function, AKAP79/150 (human79/rodent150; also known as AKAP5) plays a pivotal role in synaptic strength. This protein acts as a molecular platform, allowing the formation of complexes controlling the phosphorylation state of postsynaptic glutamate AMPARs and L-type voltage-gated Ca^2+^ channels (LTCCs) [74,106,107,108,109]. AKAP79/150-based complexes situated in the proximity of these receptors can affect both LTP and LTD, and the strength of individual synapses [110,111,112]. AKAP79/150 is essential in coordinating the antagonistic actions of PKA and CaN that regulate AMPAR levels within the post synaptic density (PSD). Specifically, phosphorylation of the AMPAR subunit GluR1 at S845 fosters its postsynaptic accumulation during the initial phase of LTP [107], whereas its dephosphorylation results in increased AMPAR endocytosis during LTD [113,114,115]. AKAP79/150 is targeted to the PSD via direct interaction with the plasma membrane [116] or by binding to N-cadherin [117] and F-actin [118]. Within the PSD, AKAP79/150 is dynamically connected to NMDARs and AMPARs by the scaffolding proteins PSD-95 and SAP97, respectively [119,120]. Thanks to its connection to these scaffolds, AKAP79/150 has also a structural function within the PSD. Indeed, it participates in the regulation of the synaptic size and strength, which ultimately underlie cognitive process [121].

### 2.5. Calcium (Ca^2+^) Compartmentalization

Cyclic AMP is not the only second messenger to achieve functional pleiotropy through compartmentalization. For this reason, we will also provide a concise overview of the role that compartmentalization of the other major second messenger, Ca^2+^, may have in ageing and age-related NDs. Under physiological conditions, cytosolic Ca^2+^ levels are maintained to nanomolar levels by the concerted activity of Ca^2+^ transporters located at the plasma membrane (PM) and the Ca^2+^-storing organelles, the main of which is the endoplasmic/sarcoplasmic reticulum (ER/SR); nevertheless, other organelles such as mitochondria [122], Golgi apparatus [123], endosomes and lysosomes [124] actively participate in shaping Ca^2+^ signals. Changes in the concentration patterns of Ca^2+^ are rapidly sensed by the cell and decoded to finely regulate a wide range of physiological events, such as release of neurotransmitters, cellular proliferation, muscle contraction and regulation of cell death [125].

### 2.6. Alterations of cAMP Compartmentalization in Aging

#### 2.6.1. ACs

The correlation between ACs and ageing was firstly demonstrated in the heart, where AC5 and AC6 are major players in the sympathetic regulation of cardiac function [50,52]. In addition, other AC isoforms such as AC8 have been linked to age-dependent myocardial dysfunction and age-dependent cardiac remodeling [126]. Stimulation of beta-adrenergic receptors (β-ARs) results in increased heart rate, contractility, and blood pressure. A common consequence of ageing is the decline in heart function, which is usually associated with altered β-AR signalling [127]. Notably, the age-related functional and structural changes, which include increased cardiomyocyte size and myocardial thickness [128], resemble those observed in heart failure (HF) [129]. At the molecular level, the observed reduction of β-AR density [130] and decreased β-AR responsiveness could be explained by disturbances of the cAMP/PKA pathway compartmentalization, which result from altered β-AR/AC coupling [131,132,133,134]. Interestingly, both in ageing and chronic conditions such as HF, the decrease in cardiac outcome is compensated by increased sympathetic activity, that, however, can eventually lead to cardiac remodeling and increased cardiomyocyte apoptosis with adverse patient outcomes [135,136,137]. Notably, β-AR antagonists have protective effects on patients of HF with decreased ejection fraction (EF) but not, paradoxically, on those with preserved EF [138]. Taken together, these data showcase the complexity of β-AR signalling and its pharmacological targeting, and the need for alternative therapeutic strategies targeting different components of the signalling cascade, as, for example, ACs.

Animal studies have suggested that, during ageing, the lack of AC5 is advantageous in several respects. AC5 is a major regulator of cardiac inotropy and chronotropy, however it could also have adverse effects for the heart. In fact, in mouse models, cardiac-specific overexpression of AC5 enhanced basal cardiac performance [139], but also decreased the cardiac resistance to prolonged stress [140]. The adverse effects of AC5 overexpression are attributed to a pathway that involves the SIRT1/FoxO3a axis [141]. Indeed, in AC5 overexpression models, prolonged catecholamine stress results in SIRT1 and FoxO3a inhibition that in turn decrease MnSOD transcription resulting in severe cardiomyopathy (Figure 2 left panel). Conversely, AC5 disruption protects the heart from the cardiomyopathy induced by chronic pressure overload and catecholamine stress, and leads to better exercise performance [142,143,144,145,146], most likely due to preserved SIRT1 and FoxO3 activity and MnSOD levels [140]. In addition, proteomic analysis in mice lacking AC5, which show increased lifespan and stress resistance, evidenced significant activation of the Raf/MEK/ERK signalling cascade, which in turn upregulated MnSOD levels during ageing [145] (Figure 2 right panel). MnSOD protects from oxidative stress and promotes longevity and its upregulation could account for the 30% lifespan increase in AC5 KO compared to wild type (wt) [145]. Interestingly, this appears to be a common mechanism among longevity models, including the one due to caloric restriction. Indeed, both these models have similar gene regulation in various tissues, and they are protected from age-dependent cardiomyopathy as well as diabetes and obesity [147]. Ageing AC5 KO mice are protected also against osteoporosis, and display improved exercise capacity [142]. From a mechanistic point of view, AC5 deletion, in addition to offering protection from oxidative stress, appears to mimic the effects of exercise training by up-regulating the SIRT1/PGC-1α pathway and nitric oxide signalling [148]. Untrained AC5 KO animals present lower levels of sympathetic tone compared to wt animals, however they show better exercise capacity [148]. Moreover, ageing is a risk factor for spontaneous onset of cancers as suggested by the finding that AC5 KO animals had lower incidence of common age-related neoplasms, were protected from mammary tumor development and displayed decreased melanoma growth. These studies were further corroborated by independent investigations on the use of an FDA approved pharmacological inhibitor of AC5 as a possible pharmacological approach for cancer treatment and prevention [149]. All together these beneficial effects suggest that lack of AC5 promotes healthy ageing, most likely regulating the ability of the cells to cope with oxidative stress [140].

An additional major cardiac AC isoform is AC6. While AC5 and AC6 share high sequence similarity (up to 65%) and are both inhibited by sub-micromolar concentrations of Ca^2+^, they display no functional overlap [150,151]. In fact, AC6 inhibition or ablation produces none of the beneficial phenotypes observed in AC5 KO animals. On the contrary, depletion of AC6 impairs cAMP production and Ca^2+^ handling, which lead to decreased Left Ventricular (LV) function that, interestingly, is more severe in males [152,153]. Cardiac overexpression of AC6 improves the function of ageing hearts [152] and shows beneficial effects in cardiomyopathy, myocardial infarction and HF [154,155,156]. In line with these findings, increased AC6 expression in the hearts of mice with severe congestive HF improved LV systolic and diastolic function and reduced cardiomyocyte apoptosis [157]; it did not protect, however, from chronic pressure overload [158]. The precise molecular mechanisms underlying the differences between the functional outcomes of AC5 and AC6 remain to be established; however, given their high similarity, it is tempting to speculate that their subcellular distribution may be the determining factor. In fact, AC6 was proposed to localize in non-caveolin fractions while AC5 was found principally in caveolin-rich fractions in vascular smooth muscle cells [159]. However, whether this differential localization is functionally significant remains to be established.

A recent study linked AC3 mutations to age-related increase in adiposity. In fact, AC3 KO mice develop age-dependent obesity, and recessive mutations in the human gene encoding AC3 were identified in two families with severe obesity. These data suggest a crucial involvement of AC3 in the control of age-dependent weight regulation, pointing to this enzyme as a promising pharmacological target [160].

In the ageing brain, synaptic function gradually deteriorates, causing the decline of synaptic plasticity and consequent memory impairment. In ageing brains and associated diseases, a marked decrease in the cAMP levels (with specific exceptions in prefrontal cortex and in hippocampus of brains affected by Huntington’s Disease (HD)) is frequently observed [9,161,162,163]. Lower cAMP levels could signify decreased production or increased degradation of the messenger, therefore the assessment of AC and PDE levels and activity is crucial for understanding the signalling events and their role in ageing and associated diseases.

In humans, a reduction in AC activity was observed both in ageing and diseased brains [9,164,165,166]. Animal studies corroborated these findings to a good extent, consistently reporting a reduction in the basal or stimulated levels of cAMP during ageing [163,167,168,169,170]. Activation of the cAMP/PKA axis by Ca^2+^ is a key pathway for synaptic plasticity and memory [171,172], and, during ageing, the crosstalk between Ca^2+^ and cAMP can be less effective, resulting in decreased Ca^2+^-stimulated ACs activation (especially in the hippocampus) with grave consequences [173]. AC1 and AC8 are the major Ca^2+^-stimulated ACs in the brain. Increasing the expression of AC1 (AC1 Tg mice) selectively enhanced recognition memory in young mice without any effect on fear or spatial memory. Surprisingly, aged AC1 Tg mice had poorer spatial memory compared to aged-matched wt controls [173]. Based on these findings it was proposed that the age-dependent decrease in Ca^2+^-stimulated ACs activity may be an adaptive mechanism to maintain spatial memory formation [173]. Interestingly, this hypothesis was also supported by studies showing that Ca^2+^/CaM-sensitive ACs likely contribute to long-term memory formation in humans and by postmortem studies on Alzheimer’s brains finding that AC1 levels were significantly decreased [174].

#### 2.6.2. PDEs

The reported levels of PDEs during ageing are not uniform, often region-specific and sometimes contradictory [9]. For instance, during ageing, an increase in PDE expression is detected in hippocampus, whereas a decrease is observed in the cortex. In the striatum and cerebellum the situation is even more variable, with PDE levels increasing or decreasing depending on the isoform type. In regards of activity (which sometimes is decoupled from the absolute expression levels), in hippocampus, cortex, striatum, and cerebellum the overall PDE activity increases with ageing [9].

Given the broad spectrum of diseases that present altered PDE levels and/or activity, PDEs became the focus of therapeutic endeavors targeting ageing-related disorders. However, despite the interest in PDEs as drug targets, the possible changes in the expression of PDE isoforms across the lifespan is poorly characterized. A unique comprehensive study on region-specific (17 PDE isoforms in 4 brain regions) and age-related PDE expression patterns [175] revealed that patterns in rodents are largely consistent with those in humans and that a select minority of PDE isoforms exhibit brain region-specific changes in expression from early to late adulthood. The most robust age-related changes in PDE mRNA and protein expression were increased PDE11A and PDE8A in hippocampus and PDE8A and PDE1C in the striatum. Of interest, PDE11A and PDE8A display particularly enriched expression in the brain versus the periphery. These age-related changes in PDE expression may reflect an attempt to restore the balance of cyclic nucleotide signaling in ageing [175]. On the other hand, dysregulation of the cAMP/PKA pathway in ageing can also be due to mis-compartmentalization of PDEs and their consequent reduction in dendrites [176,177]. Indeed, PDE inhibition phenocopies or exacerbates neuronal ageing in young or old monkeys respectively [177,178].

#### 2.6.3. PKA

Under normal non-pathologic conditions, the role of PKA compartmentalization in the brain is showcased by its involvement in synaptic plasticity. This process depends on the activity of N-methyl-D-aspartate (NMDA) receptors and α-amino-3-hydroxy-5-methyl-4-isoxazolepropionic acid (AMPA) receptors, both for strengthening (through long term potentiation, LTP) and weakening (through long term depression, LTD) synaptic efficiency. PKA is directly involved in these processes as it regulates both the number and biophysical status of AMPA receptors (AMPARs) in the PSD regions [179] (see Figure 3). In addition to synaptic plasticity, PKA impinges on synaptic maintenance and memory consolidation through its target CREB that plays a fundamental role in the conversion of short-term to long-term memory [10]. Upon PKA-dependent phosphorylation at S133, CREB assembles with its cofactors in a multiproteic complex at the level of CRE sequences in the promoters of genes involved in synaptic plasticity, regulating their transcription [180,181]. In addition, CREB can affect memory and other cognitive processes indirectly, by regulating adult hippocampal neurogenesis [182]. The role of the PKA/CREB axis in the nervous system is further testified by increasing evidence connecting changes in this pathway to deficits in brain function typical of ageing and age-related diseases, both in human patients and in animal models [183].

During ageing, PKA signalling in the brain undergoes profound variations that appear to be region-specific. For instance, in aged hippocampus, PKA activity is reduced, which leads to defective CREB signalling and impaired memory consolidation [184,185]. The opposite trend emerges in the prefrontal cortex, where the cAMP/PKA axis appears to become overactive, impairing working memory capacity [178]. Information on the specific PKA subunits affected during ageing can be gathered by animal studies. Indeed, thanks to a number of mouse models lacking specific PKA subunits [186], it became evident that, although different types of C and R subunits are not redundant and their ablation gives rise to specific defects, it is not unusual for compensation events to occur.

Among the PKA catalytic subunits, ablation of Cα results in increased prenatal mortality, with the 1/3 of the homozygote mice that survive presenting severe growth defects [187], attributed to low levels of epidermal growth factor receptor [188]. On the other hand, ablation of Cβ resulted in decreased basal PKA activity in the amygdala, hippocampus, and cortex, which was connected to learning deficiencies. These findings suggested a role of Cβ-mediated signalling in memory, albeit with some strain-specific genetic variability in the phenotypes [189]. Later work showed that overfed Cβ KO mice remained lean and were protected against steatosis, dyslipoproteinemia and insulin resistance. Interestingly, females were more protected than males, suggesting sex-dependent variations [190]. A recent report proposed that the increased glucose tolerance and metabolic rate of Cβ KO mice could depend on increased sympathetic outflow stemming from alterations in presympathetic neuronal activity [191]. Indeed, Cβ activity inhibits norepinephrine (NE) release from noradrenergic neurons; therefore, its ablation could result in increased NE secretion [191]. This mechanism however, would not fit well the cardioprotective effect observed in Cβ KO animals [192], as it is well established that prolonged NE increases can be detrimental for the heart in a PKA-dependent manner [193]. Nevertheless, sympathetic NE release could have an effect similar to temperature preconditioning, implicating slight increases in PKA activity with cardioprotective effects [194].

In addition to depletion of the PKA catalytic subunits, a number of animal models targeting the regulatory subunits of PKA have been generated. Mice lacking RIα die during gestation due to failure of normal mesodermal development, most likely because of excess PKA activity [195]. Contrary to RIα, ablation of RIIβ and RIIα results in milder phenotypes. RIIα KO mice are resistant to diet-induced obesity, present higher glucose tolerance and are protected from hepatic steatosis. Interestingly, these characteristics, as with the Cβ KO animals, are more pronounced in females [196]. Similarly to RIIα, deletion of RIIβ results in lean animals that are resistant to diet-induced obesity and fatty liver development. RIIβ KO animals display a compensatory increase of RIα, which results in overactive PKA due to the higher affinity of these subunits for cAMP. As a result of increased PKA activity, RIIβ KO animals display higher uncoupling protein induction and elevated metabolic rates [197]. Subsequent studies suggested that PKA holoenzymes containing RIα subunits could not functionally substitute the RIIβ-based PKA in its inhibitory action on the insulin-dependent activation of mitogen-activated protein kinase (MAPK) [198]. On the contrary, in RIIβ KO mice the abundance of RIα-based PKA resulted in increased basal lipolysis that was scarcely induced, however, by β-AR stimulation. These differences could be attributed to the altered compartmentalization of RIα-based versus RIIβ-based PKA, which is required for the efficient transduction of signals that modulate lipolysis [199]. In fact, it is well established that only a small number of AKAPs can engage both RI and RII subunits, with the majority of these tethers binding selectively RI- or RII-based PKA holoenzymes. Based on this, compensatory events between RI and RII could reconstitute the activity of PKA however most likely cannot fully restore the subcellular compartmentalization of PKA. Interestingly, RIIβ ablation resulted in increased lifespan and was beneficial to several age-related phenomena such as obesity, weight loss at end of life, cardiac hypertrophy, insulin resistance and the incidence and severity of age-related pathologies, albeit these effects were more pronounced in males [190]. The anti-ageing effects of PKA ablation thus seem to vary depending on the gender, as well as on model organisms. For instance, PKA *deficiency* extends lifespan in yeast [200], while on the contrary, PKA *activation* results in longevity in both *C. elegans* [201] and *D. melanogaster* [202]. Taken together, these data suggest a pivotal role for the cAMP/PKA pathway in ageing, making this cascade an appealing target for age-related disease research.

#### 2.6.4. AKAPs

Despite the importance of compartmentalization in the functional outcomes of the cAMP signalling pathway, the effects of altered PKA subcellular targeting on ageing and neurodegeneration have been scarcely investigated. Indirect evidences on this matter can be gathered by studying the effects of AKAPs, the proteins responsible for the subcellular distribution of PKA. However, while AKAPs have been implicated in the pathophysiology of many human diseases [203,204], their role in age-related pathologies has been less appreciated and investigated, and only a limited number of studies have addressed the possible connection between altered cAMP/PKA signals and AKAP function in ageing and age-related cognitive decline. A well-characterized cascade revolves around AKAP79/150, that is involved in the coordination of NMDAR-independent LTP, in which stimulation of β_2_-ARs results in increased PKA dependent phosphorylation of AMPAR GluR1 at Ser-845 [205], and of CaV_1.2_ LTCCs [206,207]. The CaV_1.2_ channel controls LTP, learning performance and the formation of spatial memory [208]. The temporal signature and intensity of CaV_1.2_-depedent Ca^2+^ signals are regulated by a multiproteic complex assembled by AKAP79/150, containing PKA, β_2_-ARs [209,210], AMPARs, CaN and an adenylyl cyclase [211,212]. Thanks to the scaffolding role of AKAP79/150, CaV_1.2_ phosphorylation and consequently function are finely regulated: β_2_-AR stimulation activates AKAP79/150–targeted PKA which phosphorylates CaV_1.2_ on S1928 to enhance its channel activity, whereas, on the contrary, Ca^2+^/CaM-activated CaN counteracts PKA-mediated enhancement, as a negative feedback regulation to hinder Ca^2+^ entry [211,213] (Figure 3).

Animal models lacking AKAP79/150 (AKAP79/150 KO) or expressing a mutant version unable to bind PKA (AKAP150 ΔPKA) result in flawed PKA recruitment and distribution within the PSD, and consequently display learning and memory defects [106,107,214]. The perturbations in the plasticity of excitatory hippocampal synapses observed during ageing can be a consequence of deregulation in AMPAR trafficking [215], which is modulated by CaMKII [216,217,218] and CaN [219,220], and of alterations in CaV_1.2_ function, that are found in several neurologic disorders including PD and AD [221].

#### 2.6.5. Ca^2+^

In the excitable cells of the brain, Ca^2+^ regulates a plethora of molecular processes. Interestingly, during ageing, adjustments in Ca^2+^ handling act as a compensatory mechanism against age-related neuronal dysfunction. For instance, in aged peripheral autonomic neurons, reduction in the ability of ER to uptake Ca^2+^, due to decreased Sarco-Endoplasmic Reticulum Calcium ATPase (SERCA) activity, is compensated by increased Ca^2+^ extrusion and buffering by the Plasma Membrane Calcium ATPase (PMCA) and by mitochondria, respectively [222]. However, these compensatory mechanisms are not resolutive and can eventually became harmful. For example, prolonged increases in intramitochondrial Ca^2+^ can trigger pathological responses that eventually lead to cell death. Impaired Ca^2+^ handling is a common age-related feature also in other neuron populations, including those of the basal forebrain [223,224]. In ageing rat neurons for example, despite the fact that subtle increases in Ca^2+^ influx through voltage-gated Ca^2+^ channels are buffered by Ca^2+^-binding proteins and basal [Ca^2+^] appears constant, synaptic transmission is impaired leading to cognitive deficiencies. In an in vitro model of aged rat hippocampal neurons, neuronal ageing is associated to increased transfer of Ca^2+^ from ER to mitochondria and impairment of store-operated Ca^2+^ entry (SOCE), a Ca^2+^ entry pathway related to memory storage [225,226]. The validity of this experimental model was demonstrated by the finding that altered expression of Orai1/Stim1 (involved in SOCE) and of Ca^2+^ channels, such as the NMDARs, the inositol triphosphate receptors (IP3Rs) and the mitochondrial Ca^2+^ uniporter (MCU), mirrors the events observed in ageing neurons in vivo [227]. In brain tissues from aged monkeys, both the mitochondria Ca^2+^ buffering capacity as well as their bioenergetic potential are reduced, and are associated to motor decline [228]. The use of genetically encoded Ca^2+^ sensors demonstrated an elevation in Ca^2+^ influx in the presynaptic terminals of aged hippocampal neurons that resulted in increased basal Ca^2+^ levels. Consequently, these animals displayed alterations in hippocampal-dependent behavioral tasks such as spatial memory [229]. A key player in neuronal Ca^2+^ dynamics appears to be the FK506-binding protein 12.6/1b (FKBP1b). This protein negatively regulates both the ER Ca^2+^ release through ryanodine receptors (RyRs) and its influx through LTCCs [230]. FKBP1b was found downregulated in hippocampal cells of aged rats and, interestingly, when its expression was restored, Ca^2+^ dysregulation was repaired, with consequent reduced cognitive impairment and ameliorated memory in ageing animal models [231,232].

## 3. Alterations of cAMP Compartmentalization in Neurodegenerative Diseases

### 3.1. Alzheimer’s Disease (AD)

While a small percentage of AD patients develop the disease early due to genetic causes, for the common late onset Alzheimer’s disease (LOAD), a major risk factor is ageing [233]. In AD, cAMP levels are found often decreased, but also increased or unchanged, depending on specific brain regions (reviewed in [9]). However, in many studies, PKA activity appears to be suppressed [9,19,234]. Tau is an unfolded, highly soluble protein involved in tubulin assembly, microtubules stabilization and stress granule axonal trafficking [235,236]. In a mouse model of tauopathy, activation of PKA early in the disease was shown to be beneficial in attenuating Tau-driven proteasome dysfunction, leading to lower levels of aggregated Tau and improved cognitive performance [237,238], underlying the fundamental role of PKA in phosphorylating proteasome subunits. Another important target of the cAMP/PKA axis, involved in AD pathogenesis, is CREB (reviewed in [239]). A drop in the phosphorylation of CREB leads to the decreased expression of many genes, including other transcription factors such as PGC1α (with implications for mitochondrial biogenesis), and NFkB (driving inflammatory response) [19,240,241,242,243,244].

The decrease in CREB activity could be counteracted by PDE inhibition, which increased the interest on the therapeutic potential of targeting these enzymes. For example, a nonspecific PDE inhibitor, propentofylline, had positive outcomes in animal models and several phase III trials, improving many AD phenotypes [19,245,246]. Notwithstanding its beneficial effects, which persisted also after treatment cessation, the use of this drug in human patients has been abandoned [19]. In fact, the large number of PDE isoforms expressed in human brain, together with their non-uniform expression patterns, represents a significant obstacle in the therapeutic exploitation of these enzymes [247]; therefore, the main challenge consists in the development of highly specific molecules, targeting single PDE isoforms [19]. For example, a number of clinical trials are currently testing PDE4 inhibitors for the treatment of LOAD [248] (https://clinicaltrials.gov). Indeed, the PDE4-specific inhibitor rolipram has been considered as a promising treatment for NDs [249,250], and the PDE4D isoform, involved in the regulation of memory, is found upregulated in AD, suggesting a role in memory loss [251]. Specific PDE4 inhibitors such as rolipram and roflumilast, but also the similarly acting resveratrol [252,253], could be used to increase cAMP levels and modify the progress of AD [19,247,252]. Resveratrol-dependent cAMP elevation enhances the deacetylase activity of Sirt1 [62] through the activation of the EPAC1-AMPK-Sirt1 axis [252]. In addition, both rolipram and resveratrol indirectly drive mitochondrial biogenesis [252] and can prevent aged-related mitochondrial dysfunction [19].

In opposition to the beneficial effect that PDE inhibition can have on the PKA-dependent phosphorylation of CREB and of the proteasome, it has been found that, in brains of LOAD patients and AD mouse models, RyRs are leaky, in part owing to their excessive PKA-dependent phosphorylation [254,255]. Taken together, the above cited examples highlight the relevance of the correct compartmentalization of the cAMP/PKA signals and underline the importance of identifying the PDEs responsible for each domain and developing highly specific PDE inhibitors [66,256,257] that could exert a therapeutic action without disrupting essential cAMP/PKA microdomains. A further cautionary note should be made on the use of mice as sole animal model; indeed, mice appear as good models for the study of early-onset genetic AD, but, due to their brain anatomy and short lifespan, are less optimal for studying the age-related LOAD [257]. For instance, in mouse models (but not in human LOAD, see below), the production of soluble Aβ oligomers is sufficient to initiate the cognitive deficits characteristic of the disease, before the insurgence of amyloid plaques [258]. Several lines of evidence suggest that chronic CaN activation may be involved in both the cognitive and degenerative effects of Aβ [259,260,261,262]. Activation of AKAP79/150-tethered CaN by Aβ-triggered increases in LTCC-dependent Ca^2+^ influx leads to the dephosphorylation of multiple NFAT phosphorylation sites and regulates NFAT signalling [263] (see Figure 3). Indeed, persistent CaN activation leads to down-regulation of NFAT targets including synaptic genes, resembling the transcriptional profiles encountered in human AD brain [264]. However, despite the well-established role of AKAP79/150 in CaN targeting in the PSD, there is no direct evidence that uncoupling of CaN from the AKAP79/150 signalling complex can contrast Aβ-dependent synaptic dysfunction.

On the contrary, in human brains affected by LOAD, the Tau aggregation precedes amyloid plaques deposition [265]. In AD, Tau becomes hyperphosphorylated and unable to bind the microtubules, and aggregates forming intracellular NFTs, both in neurons [22] and astrocytes [23]. A very recent study highlighted that changes in the phosphorylation state of Tau are associated with variations in structural, metabolic, neurodegenerative and clinical markers of disease. Interestingly, some of these changes begin simultaneously to the initial increases in β-amyloid aggregates, two decades before the development of Tau aggregates and the insurgence of the pathology [266]. In addition, amyloid plaques are found in the brains of elderly individuals that do not present cognitive impairments, while on the contrary, cognitive symptoms seem more likely to occur in brains presenting Tau tangles [267,268]. In line with these considerations, strategies to reduce Aβ have limited benefit to patients in early stages of LOAD [269].

In neurons, the physiological role of Tau is to stabilize the microtubular scaffold of axons and dendrites in response to regulatory events that are translated by cycles of phosphorylation/dephosphorylation, involving several kinases including PKA [270]. This dynamic equilibrium is lost in pathology, resulting in hyperphosphorylated Tau. As a consequence, microtubules are destabilized and collapse, disabling intracellular transport [24,25]. Thus, dysregulated cAMP/PKA, but also Ca^2+^ signalling play major roles in the pathways that lead to Tau hyperphosphorylation and to cognitive deficits during ageing [178,271,272,273], priming Tau for hyperphosphorylation by both wt and truncated forms of glycogen synthase kinase 3β (GSK3β), the latter associated with Tau pathology in LOAD [274]. This process appears connected to GSKIP (GSK3β-Interacting Protein), a small cytosolic AKAP [275] that directly interacts with both GSK3β and PKA, coordinating their actions. The involvement of a cAMP/GSKIP/GSK3β/PKA/Tau axis in Tau phosphorylation was recently studied in SHSY5Y cells and confirmed in cerebrospinal fluid and pluripotent stem cells of AD patients [276]. The authors demonstrated the existence of a molecular complex consisting of GSKIP, RII-based PKA, GSK3β and Tau (tethered to GSK3β) that enhances the cAMP/PKA signalling, resulting in increased Tau phosphorylation at distinct PKA-dependent phosphorylation sites (Ser214, Ser262, and Ser409) during AD pathogenesis [276]. Among several Tau phosphorylation sites associated with neurodegeneration [277,278], Ser262 hyperphosphorylation significantly reduces the affinity of Tau for microtubules [279] and is associated with the initial steps of AD [280]. Interestingly, a high-throughput siRNA-based screening investigating the proteins involved in Tau Ser262 phosphorylation identified another AKAP (AKAP13) as a protein able to induce S262 phosphorylation [281]. While this study suggested that AKAP13 could form a multiproteic signalling complex to regulate Tau phosphorylation and microtubule dynamics in response to extracellular stimuli, unequivocal evidence for a direct connection between AKAP13 and Tau phosphorylation is lacking.

Hyperphosphorylated Tau is prone to aggregation, eventually leading to NFTs [282], which develop in the entorhinal cortex (ERC) before the first symptoms of the disease, and extend to the deep layers of the ERC, the hippocampus and the dorsolateral prefrontal cortex (dlPFC) during the early signs of the pathology. On the contrary, the primary visual and auditory cortex remain unaffected until the very late stages of the disease [265]. Interestingly, the role of cAMP in dlPFC and ERC is quite different from its function in the visual cortex. In fact, neurons of the visual cortex rely mostly on AMPARs, and respond positively to cAMP/PKA signalling by increasing their sensory-evoked firing [283]. On the contrary, neurons in dlPFC can maintain firing even in the absence of sensory stimulation and rely on stimulation by NMDARs [284]. In these cells, cAMP reduces neuronal firing [285], and, consequently, synaptic strength. The cAMP-dependent reduction of synaptic strength in the association cortices is likely to be a physiological safety mechanism, fundamental for appropriate stress responses due to the release of catecholamines and consequent cAMP increases. However, persistent high levels of cAMP, in conjunction with high Ca^2+^, can lead to excessive PKA-dependent phosphorylation of Tau [274]. Overall, it is becoming clear that stress-induced dysregulation of signalling events is key for physiological ageing and the pathogenesis of LOAD [286,287]. These considerations highlight the need of strict regulation of the cAMP levels by PDEs [2], and of PKA activity by phosphatases [5], and could explain why stress is a risk factor especially in females, who are more prone to develop stress-induced prefrontal cortex dysfunctions [288] and LOAD [289].

Finally, it is becoming increasingly clear that also altered intracellular Ca^2+^ responses are a common feature in AD, especially in cases of familial mutations in the genes encoding APP or the γ-secretase/presenilin1 and 2 (PS1/PS2) (reviewed in [290,291]). However, the pathogenic mechanisms through which mutations in APP and PSs affect intracellular Ca^2+^ are not fully understood [292]. The “Ca^2+^ hypothesis” envisages that PS mutations result in increased ER Ca^2+^ content and excessive Ca^2+^ release in the cytosol, which affects APP processing, increases neuronal sensitization to Aβ and eventually leads to cell death [293]. In mouse models of AD, intracellular Ca^2+^ overload is caused by the deposition of β amyloid plaques in at least 20% of neurites, especially near the sites of plaque formation, underlying the fact that senile plaques are focal sources of toxicity [294]. Thus, on one hand, cytosolic Ca^2+^ overload seems responsible of structural and functional derangement of neuronal networks, while, on the other, Ca^2+^ homeostasis and compartmentalization is deeply altered by amyloid plaque deposition. In addition, dysfunctional neuronal Ca^2+^ homeostasis supports the onset of the inflammatory response in glial cells (astrocytes and microglia), accelerating the progression of the disease [295]. The involvement of deregulated ER Ca^2+^ release in the pathways that lead to the insurgence and progression of AD is still debated [296]. PSs, transmembrane proteins that support the cleavage of the APP by γ-secretase, function as SR/ER Ca^2+^ leak channels [297,298]. Genetic ablation of PSs in primary hippocampal neurons of mice reduced the expression and activity of RyRs [299]. The resulting perturbation in Ca^2+^ handling may be one of the early pathogenic events that lead to presynaptic dysfunction in AD. More recent work found that the activity of the brain-specific RyR2 isoform is altered in AD, leading to abnormalities in the metabolism of β amyloid. In particular, RyR2 undergoes post-translational modifications (PKA-dependent phosphorylation, oxidation, nitrosylation) that destabilize its macromolecular complex with FKBP1b and lead to abnormal Ca^2+^ signaling in brains of AD patients [255]. Other studies found a number of alterations in the expression patterns of RyR2 isoforms in *post-mortem* mid-temporal cortices from individuals with mild cognitive impairment and in AD brains [300], further consolidating the idea that RyR expression may reflect the onset of pathologic signalling events at early stages of the disease. Dysregulated Ca^2+^ release from the ER is coupled to mitochondrial Ca^2+^ overload and can lead to neuronal cell death [301]. Mitochondrial Ca^2+^ overload was found in APP/PS1 transgenic animals and wt littermates treated with soluble β amyloid peptide, suggesting a connection between β amyloid aggregation/deposition and mitochondrial Ca^2+^ dynamics. Interestingly, inhibition of mitochondrial Ca^2+^ uptake prevented amyloid plaque deposition in vivo [302]. In line with these findings, mitochondrial Ca^2+^ efflux was found impaired in experimental models of AD, while the NCX3 was lost in the frontal cortex of non-familial AD patients before the insurgence of neuropathology and memory decline [303].

### 3.2. Parkinson’s Disease (PD)

Age is a major risk factor for PD [304], characterized by the progressive degeneration of dopamine-producing neurons within the substantia nigra, that project to the striatum forming the nigrostriatal pathway [13]. The principal targets of dopamine are the medium spiny neurons (MSNs) in the striatum [14], that express dopamine D1 or D2 receptors (D1Rs or D2Rs). Dopaminergic neuron degeneration underlies many of the motor symptoms of PD patients [11,13].

The levels of PDE10A, which is highly expressed in the striatum [12,305,306], appear to decrease during the onset of PD, and to correlate with the progression and the severity of the symptoms [307]. Despite these evidences, PDE10-specific inhibitors were tested and found to be neuroprotective in PD animal models [308]. Notably, similar reduction of PDE10A associated with disease progression and beneficial effects of PDE10 inhibition are observed in HD [309,310]. In rodent brains, the highest levels of PDE10A are observed in the striatum and olfactory tubercle and much less in the cortex, hippocampus and cerebellar granular cells [311]. Such spatially confined expression renders unlikely the possibility that the beneficial effects for PD are due to PDE10 inhibition in non-neuronal tissues, and, at the same time, indicates PDE10A as an ideal target. Intriguingly, human mutations in PDE10A lead to loss of striatal PDE10A and are accompanied by an hyperkinetic movement disorder with onset in infancy, indicating that PDE10A plays a key role in regulating striato-cortical movement control [312].

Of note, increased cAMP levels in hyperkinetic movement disorders are strongly suggested also by the discovery of AC5-activating mutations in affected individuals [313,314]. AC5 is the most abundant AC subtype in the dorsal striatum, and it was proposed to play a role also in L-DOPA induced dyskinesia [315] (LID). L-DOPA is used to treat PD, but its prolonged use causes abnormal involuntary movements, called dyskinesia. In comparison to wt littermates, AC5 KO mice treated with L-DOPA show a reduction of LID and a decrease in the phosphorylation of ERK1/2, MSK1, and histone H3. Because of its involvement in mediating LID, AC5 has been considered as a possible therapeutic target in the treatment of LID in PD patients [315].

In MSNs, D1R stimulation activates the cAMP/PKA axis, promoting the phosphorylation of the NMDAR-NR2B subunit and the AMPAR-GluR1 (also called GluA1) subunit [316], thus regulating the excitability of striatal projection neurons and the postsynaptic targeting of AMPARs [316,317,318]. However, the dopamine-PKA axis has numerous cellular targets [316], therefore AKAP-dependent PKA compartmentalization is essential for adequate responses, and the deregulation of these complexes may be involved in the synaptic dysfunction observed in PD. This possibility is clearly suggested by the findings that the leucine-rich repeat kinase 2 (LRRK2), which has been found mutated in familiar forms of PD [319], can act as an AKAP and complex with PKA RIIβ [318,320]. Interestingly, Parisiadou and colleagues suggested that LRKK2 localizes PKA along the dendritic shaft; upon cAMP increases, PKA is activated and phosphorylates GluR1 and Cofilin within the dendritic spines [318], affecting synaptic plasticity and spine morphology. The access of PKA to the spines is crucial [321] and thus is tightly controlled by AKAPs. For instance, AKAP79/150 vehicles PKA in the vicinity of its targets within the dendritic spine, while LRRK2 and MAP2 [322] are AKAP-like proteins that tether PKA in the dendritic shaft. Upon disruption of LRRK2-PKA RIIβ binding, as in the PD-related missense mutation LRRK2 R1441C [323], PKA is likely free to translocate from shafts into the spines, resulting in increased GluR1 phosphorylation in response to D1Rs and morphological alterations [321]. Importantly, LRRK2-dependent regulation of PKA signalling is predominant during the synaptogenesis [321], a phase where PKA is crucial [324], which could explain the changes in spine morphology and altered synaptic transmission of MSNs detected in PD brains [325]. In addition to binding directly PKA, a study found that LRRK2 can complex also with AKAP8 [326], suggesting that LRRK2 can regulate the subcellular distribution of PKA also in an indirect way, through its interactions with additional AKAPs [322]. The involvement of cAMP/PKA signalling in the pathophysiology of several different LRRK2 mutants, affecting mitochondrial homeostasis [327] and autophagy [328,329], clearly suggests a fundamental role for the disruption of PKA compartmentalization in the pathogenesis of PD.

A cellular model of PD (SH-SY5Y cells with reduced expression of the endogenous mitochondrial kinase PINK1) exhibited mitochondrial fragmentation, increased mitochondria-derived superoxide, and induction of compensatory mitophagy. Mitochondrial dynamics is an important feature for the organelle function and trafficking to and from synapses [43,44] and is regulated by a small number of proteins including Drp1, a key regulator of mitochondrial fission [330,331]. In response to pro-fission stimuli, Drp1 is activated and migrates from the cytosol to the OMM; on the contrary, upon PKA-dependent phosphorylation Drp1 is inhibited, in both its GTPase activity and its translocation to the OMM [105,332,333]. Interestingly, overexpression of AKAP1, a protein that targets PKA to mitochondria, reversed the phenotypes attributed to loss of PINK1, and rescued parameters of mitochondrial respiratory dysfunction. In addition, mimicking PKA-dependent phosphorylation of Drp1 recapitulated many of the protective effects of AKAP1/PKA, indicating that redirecting endogenous PKA to mitochondria can compensate for deficiencies in PINK1 function, highlighting the importance of compartmentalized signalling networks in mitochondrial quality control and pathophysiology [89].

PKA-dependent inactivation of Drp1 was effectively enhanced by overexpression of wt AKAP1, but not of a PKA-binding deficient mutant, clearly indicating that mitochondrial fragmentation is inhibited by OMM-tethered PKA [94]. These data are in line with our recent finding that physiological cAMP elevations in both primary neonatal cardiac myocytes [5] and primary fibroblasts [4] resulted in enhanced PKA-dependent phosphorylation at the OMM but not in the cytosol and resulted in Drp1-dependent mitochondrial elongation [5]. The AKAP1/PKA signalosome appears to be protective against a variety of cellular stresses including, for instance, glutamate-induced oxidative stress [334]. Taking in consideration that oxidative stress is a major contributing factor in ageing-related NDs [335], it is tempting to consider this complex as a possible therapeutic target.

In addition to cAMP, intracellular Ca^2+^ transients and mitochondrial Ca^2+^ overload are also important hallmarks of PD (reviewed in [336,337,338]). The link between mitochondrial Ca^2+^ dysfunction and PD firstly emerged a decade ago and was consolidated by the identification of PD-causing mutations in genes connected with intracellular Ca^2+^ handling and mitochondrial function. In particular, the loss of PINK-1 causes massive mitochondrial Ca^2+^ overload due to the reduced activity of the mitochondrial NCX3 [339]. Additionally, in intact mitochondria, PINK1 directly interacts and phosphorylates the electrogenic mitochondrial Ca^2+^/H^+^ antiporter LETM1 at Thr192, reducing Ca^2+^ entry [340]. In fact, overexpression of the phosphomimetic LETM1-T192E protects PINK1-deficient neurons from cell death. The link of mitochondria to the loss of dopaminergic neurons in PINK1-Y431 mutant and PINK1^−/−^ zebrafish (*Danio rerio*) was consolidated by the finding that inhibition of the mitochondrial calcium uniporter (MCU) decreased neuronal death and ameliorated the mitochondrial respiratory chain function [341,342].

The deglycase DJ-1 is a pleiotropic protein associated with PD that is involved in ER-mitochondrial tethering and translocates to mitochondria under stress conditions (e.g., oxidative stress). DJ-1 can be found within the mitochondria-associated membranes (MAMs) in a macromolecular complex that guarantees Ca^2+^ transfer from ER to mitochondria [343,344]. In brains of DJ-1 KO mice, this complex precipitates in aggregates resulting in abnormal MAMs formation and function. Interestingly, these defects are rescued by the expression of wild-type DJ-1 but not by the expression of its PD-associated mutant L166P. Abnormalities in Ca^2+^ homeostasis are common determinants in age-related diseases as well as in neuropathological conditions, although several pieces of evidence suggest that alterations in intracellular Ca^2+^ handling do not necessarily result in deterioration of cellular function, thanks to compensatory mechanisms. Nevertheless, novel therapeutic strategies aiming at restoring dysfunctional Ca^2+^ homeostasis represent a promising route to implement the treatment of many NDs [345,346,347,348].

## 4. Final Considerations

As our understanding of the major cell signaling cascades advances, it is becoming clear that these evolutionarily conserved pathways may represent the “weakest link” in the process of ageing and age-related diseases. From one side, cellular signalling events guarantee the ability of cells to communicate with the extracellular environment and are at the basis of virtually all adaptive responses. On the other side, however, these signalling pathways require the constant maintenance of delicate equilibria between several molecular components, which comes with a cost, both of energy consumption but also, and most importantly, of system fragility. In fact, age-dependent alterations in a single component can result in suboptimal adaptive responses affecting cellular homeostasis and having a causal role in the development of age-related diseases. While our main research efforts have been concentrating on measurable biochemical changes (transcriptional, post-translational, etc.), little is known about the effects of ageing on the subcellular distribution of signalling pathways, a feature that is crucial for the proper generation of signalling networks and adaptive responses. Thanks to the continued development and improvement of methodologies that allow us to observe signalling events with great spatial definition it is only a matter of time until the effects of ageing on signalling topology, as well as the effect of disrupted topology on the ageing process, will emerge.

From a philosophical point of view, ageing divided even the most brilliant minds. In fact, while Plato considered old age a great liberator from earthbound desires and distractions, that enabled humans to achieve knowledge and good life, Aristotle detested old age as the decline that was both physical and moral. In a similar manner, in a debate held 2300 years after Plato and Aristotle, some of the foremost experts in ageing biology were unable to reach an undivided consensus of what ageing is [349]. It is clear that the road is long before fully understanding ageing and its mechanisms, however, we believe, it is safe to assume that the pressing need for discovering novel healthy-ageing interventions would have united the opinions of all, scientists and Greek philosophers.

## Figures and Tables

**Figure 1 cells-10-00464-f001:**
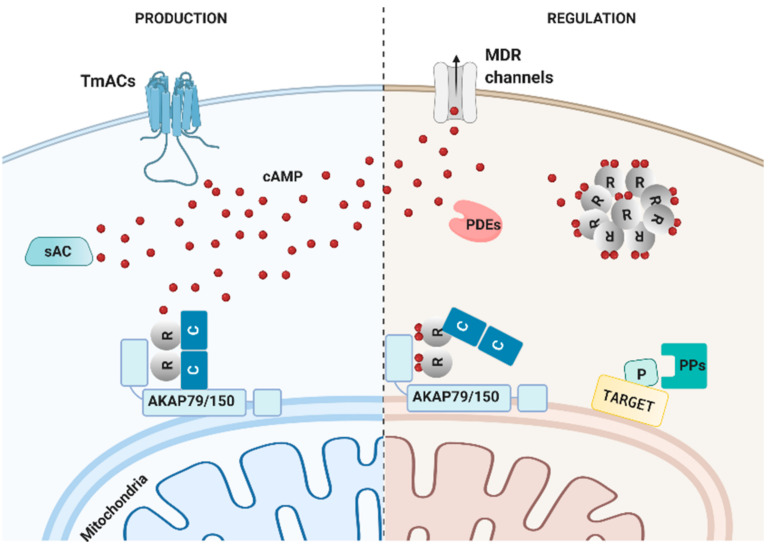
Compartmentalization mechanisms of the cAMP/PKA axis. Cyclic AMP can be produced at the plasma membrane by transmembrane adenylyl cyclases (tmACs) or intracellularly, by the soluble adenylyl cyclase (sAC). Once produced, the levels of cAMP are regulated by 3 main mechanisms: degradation by phosphodiesterases (PDEs), extrusion from the cells to the extracellular milieu by multi drug resistance channels (MDR channels) or buffering by RI alpha-constituted membraneless organelles. A final, cAMP-independent, regulatory step is exerted by the action of phosphatases (PPs) that by dephosphorylating PKA-phosphorylated targets effectively terminate the cascade. Created with BioRender.com.

**Figure 2 cells-10-00464-f002:**
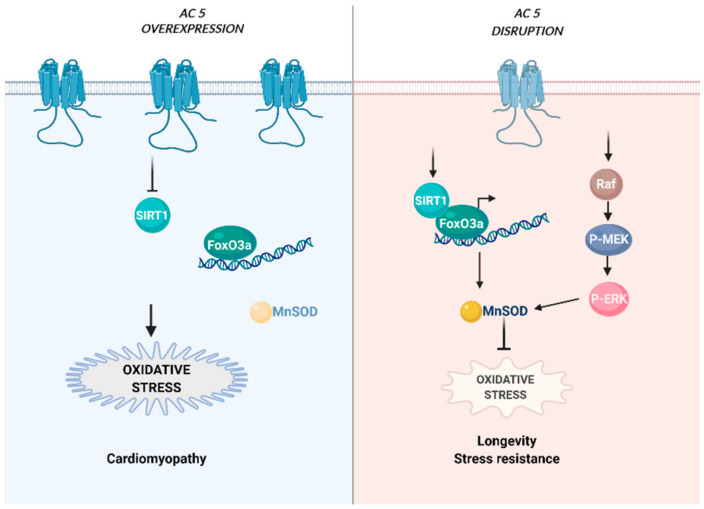
Schematic representation of the main AC5-driven pathways in ageing. (**left panel**): AC5 overexpression leads to enhanced oxidative stress through the SIRT1/FoxO3a axis. Conversely, disruption of AC5 leads to beneficial effects through the SIRT1/FoxO3 activity and the Raf/MEK/ERK signalling cascade, both impinging on the MnSOD levels (**right panel**). Created with BioRender.com.

**Figure 3 cells-10-00464-f003:**
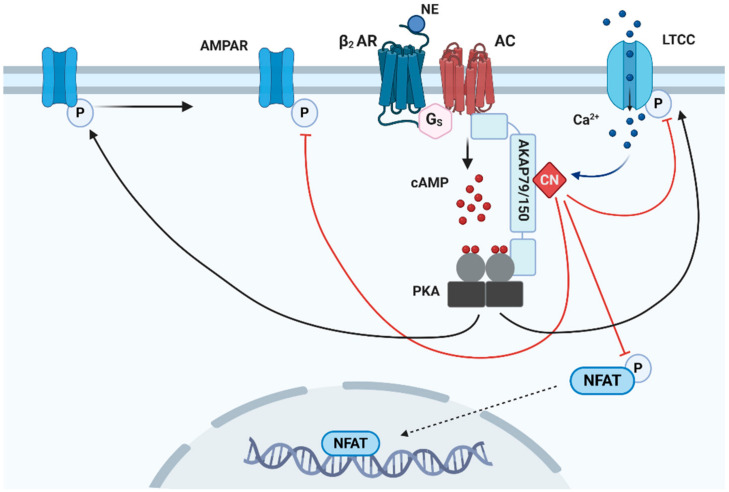
AKAP79/150 is involved in the coordination of synaptic plasticity. At the PSD, AKAP79/150 assembles a multiproteic complex containing β_2_-ARs, AMPARs, PKA, CaN and an adenylyl cyclase (likely AC5). This complex is involved in NMDAR-independent LTP, where stimulation of β_2_-ARs (by norepinephrine, NE) results in increased PKA dependent phosphorylation of AMPAR GluR1 and of CaV_1.2_ LTCCs. PKA phosphorylation promotes exocytosis, lateral diffusion and synaptic accumulation of AMPARs, and increases the opening probability of both AMPAR and LTCCs. Activation of CaN by Ca^2+^ influx through LTCC counteracts phosphorylation of both LTCC, as a negative feedback regulation to hinder Ca^2+^ entry, and AMPAR. Dephosphorylated AMPAR are removed from the synapse (not shown), leading to LTD. Moreover, activation of CaN leads to the dephosphorylation of NFAT, promoting its translocation in the nucleus, where it can act as transcriptional activator/repressor. Persistent CaN activation leads to down-regulation of NFAT targets including synaptic genes, resembling the transcriptional profiles encountered in human AD brain. A similar complex assembled by AKAP79/150 close to NMDAR (not shown) mediates LTP and contains AMPAR, PKA, CN and a Ca^2+^-activated AC (likely AC1 or AC8). Created with BioRender.com.

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
