# Peer review of "Compartmentalized Signaling in Aging and Neurodegeneration"

_cells, 2021, doi:10.3390/cells10020464_

Round 1
Reviewer 1 Report
In this review article Di Benedetto and colleagues discuss the correlation between cAMP compartmentalization and ageing, in particular focusing on age-related neurodegenerative disorders like AD and PD. This topic is rapidly expanding and the review is timely and relevant, so I recommend publication. The following comments are intented to further improve the author’s critical assessment of the field:
- The manuscript is well written however, I suggest a careful revision. Specifically, the authors should check for typos and, where possible, avoid redundancy (for example, the sentence: “an unfolded, highly soluble…” about Tau, in paragraph 5.1 is a repetition since the same concept was previously described.
- All the abbreviations must be introduced with full terminology in the first instance, not all acronyms were defined, please verify.
- In the manuscript the authors refer to Figure 1A and 1B. Please label the panels in Figure 1, they are not reported in this version.
- I understand that for a review is not easy, but the authors can try to include a general figure/cartoon depicting the overall concept, I believe that a graphical abstract could be highly helpful to non-expert readers in the field.
- I believe that after a such detailed overview a brief paragraph that critically focuses on current issues and possible future perspectives is important to conclude the manuscript on an high note.
Reviewer 2 Report
This is an exhaustive work which is essentially impossible to review because of the abundance of information. The authors cite 360 publications and try to explain essentially a significant part of cellular biology, several rather unrelated neurodegenerative diseases, and aging in their oeuvre. In my humble opinion, focusing on one disease or aging would suffice. I would also like to know more about the different proteins, e.g. in the first chapter on adenylyl cyclases, I would rather follow a certain order like AC1-X always describing domains, expression, cellular compartment (!) and important interacting proteins like the ACAP, and function before describing what goes wrong in which disease. Alternatively, instead on describing different protein classes one after another without real order, one could describe different subcellular compartments (e.g. mitochondria, ER, Golgi, cytoskeleton, nucleus, plasma membrane) and their cAMP etc. signaling. There is also only one figure. A good review lives from its figures. The figures should describe the different cellular compartments in line with the title. In summary, I think the work has its merits but should be more focused in order to become useful to the scientific community
Round 2
Reviewer 2 Report
The manuscript is much improved now and follows a more logical course. The new figures serve to illustrate the still very complex topic. Some parts seem to be formatted differently and there are still some minor orthographic mistakes e.g. extend instead of extent on page 2. The statement that PD belongs to a group of autosomal-dominant striatal diseases on page 3 is wrong. ADSD is a disease caused by mutation of a phosphodiesterase that has some features of PD. This should be corrected. Huntington is misspelled on page 5. The lack of page numbers is annoying. But overall, I think the review should now be published after correction of these minor things.
Author Response
We would like to thank the reviewer for taking the time to review the new version of our manuscript.
We addressed all the requests in the new version of the manuscript.